# The Effect of Dual-Modification by Heat-Moisture Treatment and Octenylsuccinylation on Physicochemical and Pasting Properties of Arrowroot Starch

**DOI:** 10.3390/polym15153215

**Published:** 2023-07-28

**Authors:** Herlina Marta, Ari Rismawati, Giffary Pramafisi Soeherman, Yana Cahyana, Mohamad Djali, Tri Yuliana, Dewi Sondari

**Affiliations:** 1Department of Food Technology, Faculty of Agro-Industrial Technology, Universitas Padjadjaran, Bandung 45363, Indonesia; ari20001@mail.unpad.ac.id (A.R.); y.cahyana@unpad.ac.id (Y.C.); djali@unpad.ac.id (M.D.); t.yuliana@unpad.ac.id (T.Y.); 2Department of Agroindustry Technology, Lampung State Polytechnic, Lampung 35141, Indonesia; giffarypramafisi@polinela.ac.id; 3Research Center for Biomass and Bioproducts, Cibinong Science Center, National Research and Innovation Agency, Cibinong 16911, Indonesia; dewi004@brin.go.id

**Keywords:** arrowroot starch, heat-moisture treatment, octenylsuccinilation, dual-modification, physicochemical properties, functional properties

## Abstract

Starch is widely applied in various industrial sectors, including the food industry. Starch is used as a thickener, stabilizer, or emulsifier. However, arrowroot starch generally has weaknesses, such as unstable under heating and acidic conditions, which are generally applied to processing in the food industry. Modifications were applied to improve the characteristics of native arrowroot starch. In this study, arrowroot starch was modified by heat-moisture treatment (HMT), octenylsuccinylation (OSA), and dual modification between OSA and HMT in a different sequence––namely, HMT followed by OSA, and OSA followed by HMT. This study aims to determine the effect of different modification methods on the physicochemical and functional properties of native arrowroot starch. The result shows that both single HMT and dual modification caused damage to native starch granules, such as the formation of cracks and roughness. For single OSA treatment, especially, there is no significant change in granule morphology after modification. All modification treatments did not change the crystalline type of starch but reduced the RC of native starch. Both single HMT and dual modifications (HMT-OSA, OSA-HMT) increased pasting temperature and setback, but, conversely, decreased the peak and the breakdown viscosity of native starch, whereas single OSA had the opposite trend compared with the other modifications. HMT played a greater role in increasing the thermal stability and the retrogradation ability of arrowroot starch. Both single modifications (HMT and OSA) increased the hardness and gumminess of native starch, and the opposite was true for the dual modifications. HMT had a greater effect on color characteristics, where the lightness and whiteness index of native arrowroot starch decreased. Single OSA modification increased swelling volume higher than dual modification. Both single HMT and dual modifications increased water absorption capacity and decreased the oil absorption capacity of native arrowroot starch.

## 1. Introduction

Arrowroot (*Maranta arundinacea* L.) is a tuber plant with a rhizome root, which is elongated like an arrow. Arrowroot tubers have been considered inferior commodities and have not been utilized optimally. Arrowroot tuber is one of the commodity sources of carbohydrates, which can be processed into starch or flour. Starch can be utilized in many ways, such as a food thickener, stabilizer, and emulsifier [1]. Starches that are commonly found in the market are corn starch (maize) and cassava starch (tapioca). However, there are still many potential plants that are potential sources of starch, and one of them is arrowroot tubers. 

Native arrowroot starch, however, as with starches from other sources, has several limitations, such as low thermal stability and susceptibility to acidic conditions, which are generally used in processing in the food industry [2]. To improve the characteristics of native arrowroot starch, modifications were applied. One modification method that can be used to enhance thermal stability is the heat-moisture treatment (HMT) method [3,4]. However, several studies have shown that HMT increases starch syneresis [5,6]. Dual modification technology is an alternative to overcome the weakness of HMT starch by combining HMT modification treatment with other modification treatments, such as chemical modification. This study combined both modified treatments using heat-moisture treatment and octenylsuccinilation using octenyl-succinic anhydride (OSA). Modification of OSA has advantages because it can increase the hydrophobicity of starch [7,8]. Thus, OSA-modified starch can be applied as an emulsifier or used as a fat replacer in high-fat products, resulting in decreased fat products [8,9,10].

Information regarding the modification of arrowroot starch modified by two methods, HMT and OSA, is still limited. This current study aimed to determine the physicochemical and pasting properties of native and modified starches for both single and dual modifications using HMT and OSA in the reverse sequence. HMT and OSA-modified starch is expected to be an ingredient in the manufacture of a functional food because each single modification treatment has advantages from a health perspective. For example, HMT-modified starch has very slowly digested starch, making it suitable for diabetics [11], while single OSA modification can produce starch that can replace the role of fat and could thus be used to produce low-fat products [8].

## 2. Materials and Methods

### 2.1. Materials

Arrowroot tubers (*Maranta arundinacea* L.) cultivated in Pangandaran, West Java, Indonesia were used as the raw material for starch extraction. All chemicals used for the modification process were 2-octen-1-ylsuccinic anhydride (OSA) (Sigma-Aldrich, St. Louis, MO, USA), hydrochloric acid, sodium hydroxide, ethanol 95%, acetic acid, isopropanol, AgNO_3_, and phenolphthalein, with specification analytical grade.

### 2.2. Arrowroot Starch Isolation

Arrowroot starch isolation method was followed according to Marta et al. [11] with a slight modification. The tubers were peeled and cut into small pieces, and were then further soaked in water with a tuber/water ratio of 1:5. The tubers were crushed to form a pulp using a blender (Sharp EM-121-BK, Chiba, Japan) for 2 min. The resulting slurry was squeezed using a muslin cloth. The slurry was allowed to precipitate for 18 h and decanted to separate starch and supernatant. The resulting starch was washed and centrifugated using SL 16 Centrifuge, Thermo Scientific, Waltham, MA, USA at 5000 rpm for 2–3 min to precipitate the starch. Starch washing and centrifugation were repeated 3 times in order to obtain clean starch. The resulting starch was dried in a drying oven at 50 °C for 24 h, and was then dried and sieved using a 100-mesh sieve.

### 2.3. Heat-Moisture Treated (HMT) Starch Preparation

Preparation of HMT-starch refers to Marta et al. [12]. The moisture content of arrowroot starch was adjusted to 30% (±2%) by adding distilled water, and it was then equilibrated at 4 °C for 24 h in the refrigerator. The starch was then transferred to a tightly-closed Teflon and heated at 100 °C for 8 h. The modified starch was dried in a drying oven at 50 °C for 24 h, and was then ground and sieved using a 100-mesh sieve.

### 2.4. Octenylsuccinilated (OSA) Starch Preparation

Preparation of OSA-starch refers to Marta et al. [13] with a slight modification. The starch sample was first made into a 30% (*w*/*w*) starch suspension by dissolving the starch in distilled water. The pH of the starch suspension was then adjusted to pH 8 using 1 M NaOH before the addition of 3% OSA solution (*w*/*w*). The esterification was then carried out by stirring the solution for 3 h. During the modification process, the pH of the suspension was adjusted and maintained in the range of 8.0–8.5 using 1 M NaOH and 1 M HCl. After the reaction, the starch suspension was adjusted to pH 6.5 using 1 M HCl. Starch was centrifuged at 5000 rpm for 2–3 min. The precipitated starch was washed using distilled water and then centrifuged again, and this process was repeated 2–3 times. The starch was then dried in a drying oven at 50 °C for 24 h. The dried modified starch was ground and sieved (no. 100 mesh sieve).

### 2.5. Dual Modification by HMT Followed by OSA (HMT-OSA)

The native starch of arrowroot was modified by HMT (as mentioned in Section 2.3) and followed by OSA modification, as mentioned in Section 2.4, and the obtained starch was named HMT-OSA starch.

### 2.6. Dual Modification by OSA Followed by HMT (OSA-HMT)

The native starch of arrowroot was modified by OSA (as mentioned in Section 2.4) and then followed by HMT modification, as mentioned in Section 2.3, and the obtained starch was named OSA-HMT starch.

### 2.7. Granule Morphology Observation Using Scanning Electron Microscopy (SEM)

The granular morphology of native and modified arrowroot starch was determined using scanning electron microscopy (SEM), a JEOL JSM-6360 LA at 15 kV (JEOL, Tokyo, Janpan). Mounted starch samples were coated with gold/palladium at 8–10 mA for 10–15 min under low pressure (less than 10 tors). Representative digital images of native and modified starch granules were obtained at 5000 and 10,000 magnifications.

### 2.8. Starch Crystallinity Using X-ray Diffractometer (XRD)

The crystallinity of native and modified arrowroot starch was measured using an X’Pert PRO series PW3040/60 X-ray diffractometer (Malvern Panalytical, Malvern, UK) that operated using Cu-K alpha radiation with a wavelength of 1.540 nm as an X-ray source at 40 kV and 30 mA. The diffraction angle (2θ) scanning was from 3.0° to 50.0° with a scanning rate time of 2.9 s. OriginLab Program was used to calculate the relative crystallinity of the starch samples.

### 2.9. Thermal Properties Determination Using Differential Scanning Calorimetry (DSC)

Thermal properties of starch samples were measured using DSC-Q100, TA Instruments (New Castle, DE, USA). The parameters observed were onset temperature (T_o_), peak temperature (T_p_), conclusion temperature (T_c_), and enthalpy of gelatinization (∆H). Starch was made into a slurry, with the ratio of water and starch being 3:1. The slurry was then hermetically sealed using a DuPont encapsulation press before weighing. The starch samples were heated at a rate of 5 °C/min from 20 to 100 °C.

### 2.10. Pasting Properties Determination Using Rapid Visco Analyzer (RVA)

The pasting properties of arrowroot starch were determined using an RVA StarchMaster 2, Parten Instruments. A total of 3 g of starch samples were added with 25 mL of distilled water in the RVA canister tube, and stirred in an RVA canister at 960 rpm for 10 s. RVA was set with a temperature profile, initially held at 50 °C for 1 min; the heating was from 50 to 95 °C for 3.7 min; the temperature was held at 95 °C for 2.5 min; and cooling was then achieved at 50 °C in 3.8 min and then kept at 50 °C for 2 min. The gel was then maintained at 50 °C for 2 min with constant paddle rotational speed (160 rpm) used throughout the analysis, and the total analysis time was 12 min. The pasting properties included the following parameters: pasting temperature (PT), peak viscosity (PV), hold viscosity (HV), final viscosity (FV), breakdown (BD), and setback (SB).

### 2.11. Texture Properties Evaluation Using Texture Analyzer (TA)

The texture properties of starch gels were evaluated on a TA-XT express enhanced Stable Micro System (Surrey, UK). Exponent Lite Express software was used to collect and calculate the data obtained. The gelatinized starch in the canister after the RVA measurement was poured into cylindrical plastic tubes (20 mm diameter, 40 mm deep) and then kept at 4 °C for 24 h to form a solid gel. Each gel sample in the tube was penetrated with a cylindrical probe (P36/R) at a speed of 5 mm/s to a distance of 10 mm for two penetration cycles. The texture profile curves were used to calculate hardness, adhesiveness, springiness, cohesiveness, and gumminess.

### 2.12. Functional Properties

Swelling volume and solubility of arrowroot starch were measured referring to Marta et al. [13]. A total of 0.35 g (db) of starch sample was mixed with 10 mL distilled water and put into a centrifuge tube. It was then mixed using a vortex mixer for 20 s, and the sample was then heated in a water bath at 92.5 °C for 30 min and stirred regularly. The starch sample was cooled for 1 min in ice water and centrifuged at 3500 rpm for 15 min, and then the supernatant was separated and the volume was measured. Swelling volume was calculated using the equation:(1)$$\mathrm{Swelling}\;\mathrm{volume}\;(\mathrm{mL}/g)=\frac{\mathrm{total}\;\mathrm{volume}-\mathrm{supernatant}\;\mathrm{volume}}{\mathrm{weight}\;\mathrm{of}\;\mathrm{sample}\;\left(\mathrm{db}\right)}$$

After being separated, the supernatant was dried in a drying oven for 24 h. Solubility was calculated using the equation:(2)$$\mathrm{Solubility}\;(\%)=\frac{\mathrm{weight}\;\mathrm{of}\;\mathrm{dried}\;\mathrm{supernatant}}{\mathrm{weight}\;\mathrm{of}\;\mathrm{sample}\;\left(\mathrm{db}\right)}\;\times \;100\%$$

Water absorption capacity (WAC) and oil absorption capacity (OAC) were measured referring to Marta et al. [13]. One gram (db) of the starch sample was added with 10 mL of distilled water or oil into a centrifuge tube, and then mixed using a vortex for 20 s. Samples were stored at room temperature for 1 h and then centrifuged at 3500 rpm for 30 min. The volume of the supernatant (water or oil) was measured and separated. WAC and OAC are calculated using the following equation:(3)$$\mathrm{WAC}\;(g/g)=\frac{\mathrm{volume}\;\mathrm{of}\;\mathrm{water}\;\mathrm{absorbed}}{\mathrm{weight}\;\mathrm{sample}\;\left(\mathrm{db}\right)}$$
(4)$$\mathrm{OAC}\;(g/g)=\frac{\mathrm{volume}\;\mathrm{of}\;\mathrm{oil}\;\mathrm{absorbed}}{\mathrm{weight}\;\mathrm{sample}\;\left(\mathrm{db}\right)}$$

Freeze-thaw stability or syneresis was determined by a previous method [12] with a slight modification. An aqueous starch suspension (5%) was prepared and heated at 95 °C for 30 min with constant light stirring, then cooled to room temperature in an ice water bath. A total of 20 g of aliquots of the paste was then taken and put into a centrifuge tube, and a freeze-thaw cycle was then carried out with storage at 4 °C for 24 h, frozen at −15 °C for 48 h, and then thawed at 25 °C for 3 h. The samples were then centrifuged at 3500 rpm for 15 min. The supernatant was separated from the gel. The syneresis was calculated using the following equation:(5)$$\mathrm{Syneresis}\;(\%)=\frac{\mathrm{weight}\;\mathrm{of}\;\mathrm{separated}\;\mathrm{water}}{\mathrm{weight}\;\mathrm{of}\;\mathrm{starch}\;\mathrm{paste}}\;\times \;100\%$$

### 2.13. Color Analysis Using CM-5 Spectrophotometer

The starch color was measured using a CM-5 Spectrophotometer (Konica Minolta Co., Osaka, Japan) with SpectraMagic NX version 3.0 Software. The samples were placed in a glass cell and above the light source. After that, measurements were taken at room temperature. The parameters measured were CIE-lab namely, the L*, a*, and b* values in each sample. L* value indicates the lightness (whiteness/darkness), representing dark (0) to light (100); a* value indicates the degree of red-green color ((+) redness/(−) greenness); and b* value indicates the degree of the yellow-blue color ((+) yellowness/(−) blueness). The whiteness index for native and modified starches was determined using the following equation [14]:(6)$$\mathrm{Whiteness}\;\mathrm{Index}=100\;-\sqrt{{\left(100\;-{\;L}^{*}\right)}^{2}{+a}^{*2}{+b}^{*2}\;}$$

Total color differences between starch samples were calculated using the following equation [15]:(7)$$\Delta E=\sqrt{{\left(L^{*}-{\;L}_{n}^{*}\right)}^{2}+{\left(a^{*}-{\;a}_{n}^{*}\right)}^{2}+{\left(b^{*}-b_{n}^{*}\right)}^{2}}$$

L*, a*, and b* were parameters for modified starch, and $L_{n}^{*}$,${\;a}_{n}^{*}$, and $b_{n}^{*}$ were parameters for native starch.

### 2.14. Statistical Analysis

Data are displayed as the mean ± SD. Experiments were all performed in triplicate. One-way analysis of variance (ANOVA) was used to analyze all of the data, and the Duncan test was used to compare the sample mean at a significance level of 5% (*p* < 0.05). The statistical software program IBM SPSS Statistics version 25 was used to examine all of the data.

## 3. Results

### 3.1. Granule Morphology

The granule morphology of native and modified arrowroot starches is presented in Figure 1. The granules of arrowroot starch are spherical and ellipsoid to oval-shaped. There was no damage on the surface of starch granules after OSA treatment. In HMT-modified starch, in both single and dual modifications (HMT, HMT-OSA, and OSA-HMT), the damage occurred on the surface of the granules, where the surface became rougher due to the formation of fine cracks.

### 3.2. Starch Crystallinity

The crystallinity properties of native and modified arrowroot starch are presented in Figure 2. According to the diffractograms, both native and modified arrowroot starch possessed an A-type crystalline pattern, which was indicated by several diffraction peaks at 15°, 17°, 18°, 23°, and 26° (2θ), showing that all of the treatments did not alter the crystalline pattern of the arrowroot starches. However, in terms of relative crystallinity (RC), the modified starches have lower RC compared to the native starch. The native arrowroot starch has an RC value of 30.96%, while OSA, HMT, HMT-OSA, and OSA-HMT starches have 30.25%, 28.26%, 24.88%, and 26.94% RC values, respectively. Additionally, HMT-modified starch in both single and dual modifications (HMT, HMT-OSA, and OSA-HMT) has a greater effect on RC than single OSA modification.

### 3.3. Thermal Properties

DSC is used for thermal analysis in order to determine the transition of starch crystallinity caused by heating with the following parameters: (1) T_o_, or temperature onset, is the temperature at which gelatinization begins, and is also defined as the melting temperature of the weakest crystals in starch granules; (2) T_p_, or peak temperature, represents the the endothermic peak on the DSC thermogram; (3) T_c_, or conclusion temperature, is the final temperature at which the sample is wholly gelatinized, and is sometimes known as the crystalline melting temperature (high-perfection crystalline); and (4) ΔH or enthalpy (J/g) is calculated based on the DSC endotherm, expressing the energy required to break the double helix structure during starch gelatinization [12,16].

The thermal properties of native and modified arrowroot starches are presented in Table 1. All modified starch has a lower T_o_ than native starch, except for HMT starch. T_p_ native starch decreased after modification from 50.46 °C to 39.83–44.36 °C. Both single methods (OSA and HMT) decreased T_c_, but, conversely, both dual modification methods increased the T_c_ of native starch. The temperature range (T_c_–T_o_) of modified starches was higher than native starch, except for HMT-starch, whereas the enthalpy of modified starch decreased significantly from 1117.08 (J/g) to 936.44–647.30 (J/g).

### 3.4. Pasting Properties

The viscoamylograph and pasting properties parameters of native and modified arrowroot starches are presented in Figure 3 and Table 2, respectively. Both single HMT and dual modifications (HMT-OSA, OSA-HMT) increased PT and SB of native starch, but, conversely, decreased PV and BD viscosity of native starch, whereas single OSA has the opposite trend compared with the other modification treatments.

### 3.5. Texture Properties

Texture properties parameters of native and modified arrowroot starches are presented in Table 3. Both single modifications, OSA and HMT alone, increased the gel hardness of native arrowroot starch, and the converse effect was seen for the dual modifications. All modified starches have higher adhesiveness and lower springiness and cohesiveness than native starch. Both dual modifications, HMT-OSA and OSA-HMT, decreased the gumminess of native arrowroot starch from 208.95 to 29.21–59.60.

### 3.6. Functional Properties

OSA modification, for both single OSA and dual HMT-OSA, significantly increased the SV of native starch, but we saw a converse trend for OSA-HMT. All modified starch has lower solubility and higher syneresis than native starch (Table 4). HMT starch, for both single HMT and dual modifications (HMT-OSA, OSA-HMT), increased the water absorption capacity (WAC) and decreased the oil absorption capacity (OAC) of native starch. All modification treatments increased the syneresis of native starch.

WAC describes the amount of water available for gelatinization [12]. WAC of native and modified arrowroot starches ranges from 0.83–1.60 g/g (db). The WAC of native arrowroot starch is 1.13 g/g db, which is smaller than in the previous study, where it was 1.81 g/g db [17]. Meanwhile, the ability of starch to absorb oil is called OAC, which could also represent the emulsifying properties of the starch [18]. The OAC of native and modified arrowroot starches range from 2.15–2.33 g/g db. All of the modification methods, except for OSA, decreased the OAC of native starch from 2.33 g/g db to 2.15–2.21 g/g db.

### 3.7. Color Characteristics

The color parameters of native and modified arrowroot starches are L*, a*, and b*, whiteness index, and ΔE (Table 5). HMT modification of both single HMT and dual modifications (HMT-OSA and OSA-HMT) significantly decreased the L* value of native arrowroot starch. Single OSA did not alter the a* value, whereas HMT modification of both single and dual had a different effect on the a* value of native arrowroot starch. Native and OSA-modified arrowroot starches have a negative value of a*, which indicates that the color tends towards greenness, whereas all HMT modifications have a positive value of a*, which indicates that the color tends towards redness. All modified starch has a lower whiteness index than native starch, except for OSA starch. OSA starch has the highest whiteness index among all of the samples. ∆E is a value indicating the total color difference between a modified starch and the native starch. The ∆E of modified starch ranged from 0.64 to 1.61. The higher the ∆E, the greater the color differences between modified and native starches, whereas the HMT starch has the highest ∆E among other modified starches. The color images of native and modified arrowroot starches which were captured from Spectrophotometer CM-5 are presented in Figure 4.

## 4. Discussion

### 4.1. Granule Morphology

Native arrowroot starch has a round, oval shape with a granular surface that is slightly textured without cracks, and this is in line with some of the previous studies [1,19]. The morphology of OSA starch granules did not show significant changes compared to native starch granules. This result was consistent with the study of OSA-modified Japonica rice starch [20] and OSA-modified sago starch [13], whereas the HMT caused damage to the starch granules, with cracks forming and the surface of the granules becoming rougher. Some of the previous studies also reported granule surface deformation after HMT [5,6,12], and this is due to the thermal strength, which changes the morphology of the arrowroot starch [11]. Both dual-modified starch granules (OSA-HMT and HMT-OSA) showed similar surface characteristics to the HMT starch. The surface of the granules became rougher, and there were also indentations. This indicated that thermal treatment significantly dominates the alteration of granule morphology of dual-modified arrowroot starch.

### 4.2. Crystallinity

X-ray Diffraction (XRD) is a method for assessing and quantifying long-range crystalline order in starch [21]. Figure 2 shows that native and modified starches have a similar crystallinity pattern, i.e., an A-type crystalline pattern, as indicated by several diffraction peaks at 2θ 15°, 17°, 18°, 23°, and 26° [22], which indicated that all of the modification treatments did not significantly affect the crystalline type of native arrowroot starch. Several studies have reported that native arrowroot starch has an A-type crystalline pattern [1,23]. However, this is not in agreement with another study by Nogueira et al. [24], which reported that arrowroot starch has a C-type crystalline pattern, indicating a mixture of polymorphs types A and B. The HMT showed no change in the crystalline pattern, similar to the study of Marta et al. [11] on banana starch, and a similar trend has been seen on OSA-sago starch [13].

Relative crystallinity (RC) was calculated based on the ratio of the diffraction peak area (crystalline area) to the total diffraction area [25]. Arrowroot native starch has an RC of 30.96%, which was in range with the other studies (52.84% [26] and 28.8–30.2%) [27]. All modification treatments, however, decreased the RC of native starch––that is, from 30.96% for native arrowroot starch to 30.25% for OSA-treated starch, 28.26% for HMT-treated starch, 284.88% for HMT-OSA, and 26.94% for OSA-HMT. Several studies have reported that HMT decreased the RC, such as in sago starch [13], banana starch [11,13]; mango kernel starch [28], breadfruit starch [6], rice, cassava, and pinhão starches [29]. Decreased RC in hydrothermally modified starches, both for single HMT and dual modifications (HMT, HMT-OSA, and OSA-HMT), can be associated with changes in the crystalline phase (amylopectin) of starch, where dehydration and double helix movements can disrupt starch crystallinities and change the crystal orientation of the semi-crystalline fraction to the amorphous phase [30,31,32] or possibly partial gelatinization [4,33]. The RC of OSA starches, for both single and dual modified starches (OSA, OSA-HMT, and HMT-OSA), was lower than its native counterpart, which was in line with some of the previous studies [13,34,35]. These results indicated that OSA esterification occurs in the amorphous region of starch granules and slightly changes the starch crystal structure.

### 4.3. Thermal Properties

Esterification using OSA on arrowroot starch causes a decrease in T_o_, T_p_, and ΔH compared to native starch, which is in agreement with some of the previous studies [36,37]. The introduction of OSA molecules changes the degree of hydrogen bonding, which tends to weaken the interactions between the starch macromolecules, allowing the granules to swell and melt at lower temperatures [7,38], whereas heat-moisture treatment could weaken the interaction between amylose-amylose, amylose-amylopectin, and amylose-lipid, which leads to imperfect crystal formation. This phenomenon could cause a decrease of T_c_ and ΔH in HMT-treated starch [39]. Some previous studies have reported that HMT-modified starch showed higher gelatinization parameters (T_o_, T_p_, T_c_) than native starch [12,30,40,41], whereas in this study, T_o_ and T_p_ of HMT starch were not significantly different to native starch. Both dual-modified starch HMT-OSA and OSA-HMT starches tend to have thermal characteristics that are almost similar, where the T_o_, T_p_, T_c_, and T_c_–T_o_ are not significantly different from each other, whereas ΔH of OSA-HMT starch is significantly higher than HMT-OSA starch.

### 4.4. Pasting Properties

The increase in PT and the decrease in PV on HMT starch, both for single HMT and dual modifications (HMT-OSA and OSA-HMT), could be due to the partial breakage of the ordered chain structure and the rearranging of the broken molecules, facilitated by the high temperature (100 °C) and limited moisture content (30%) during HMT. As a result, the forces of the intra-granular bonds would be augmented, and the linkages between starch chains would be strengthened. The HMT-treated starch samples needed greater heat for structural breakdown and paste production, resulting in a lower paste viscosity [4]. On the other hand, an increase in PV was observed in single OSA-treated starch. According to Bajaj et al. [8], substituting bulky octenyl groups could decrease the inter- and intramolecular bond between starch granules, resulting in limited incorporation of water into starch molecules. When OSA groups are substituted, the starch granule is destroyed, which enhances swelling and raises PV. The hydrophobic properties of OSA also increased the viscosity of starch. This characteristic was discovered advantageous for making mayonnaise with improved emulsion stability [8].

The significant difference in PV between HMT-OSA and OSA-HMT-treated starch is because HMT facilitated this OSA particle to attack more of the –OH group in starch. According to Park et al. [42], HMT before cross-linking could increase the phosphate content of the starch, showing that the HMT facilitates the incorporation of the cross-linking agent to react more with starch granules. This occurred in the octenyl group as well. The OSA modification increased the BD, which was in line with some of the previous studies [8,43], whereas the other modifications significantly decreased the BD of the native starches. The lower BD indicated the higher thermal stability of starch. All HMT modifications, both single and dual modifications, have a higher SB than native and single OSA starch, which indicates that HMT increases the ability of starch to retrograde.

From the industrial point of view, the modification of arrowroot starch could give arrowroot more value, as the starch could be used more in the food industry. HMT and dual-modified starches (OSA-HMT and HMT-OSA starcher) have a higher thermal stability than native starch, as shown by the lower breakdown viscosity of the starches. Starch with good thermal stability could be used as a thickening agent for food products that need to be sterilized, e.g., sauce, paste, etc. [39], whereas OSA starch has a lower ability to retrograde, which indicates that the starch may be used as an ingredient in baby food and baked goods (as it can inhibit staling on bread).

### 4.5. Texture Properties

Texture parameters in starch have an important role because starch can be a texturizer agent, such as a thickening and gelling agent. TPA (texture profile analysis) is used to observe texture parameters, such as hardness, adhesiveness, springiness, cohesiveness, and gumminess [44]. Gel hardness or hardness is related to the strength of the gel network, and changes in hardness are related to the effect of swelling granules and amylose content [44,45], and hardness increases with increasing amylose content. The hardness of HMT and OSA-modified starches were higher than native starch, whereas the amylose content of native starch was lower than both HMT and OSA starch. The amylose content for native, HMT, and OSA starches are 29.15%db, 36.01%db, and 33.44%db, respectively. The gel network structure depends on the quantity and intensity of hydrogen bonds formed between the amylose chains [46]. Conversely, the hardness of dual-modified starch (OSA-HMT and HMT-OSA starch) decreased very sharply compared to native starch, and it was not in line with the amylose content of the dual-modified starches, where the hardness decreased with increasing amylose content. The amylose content of HMT-OSA and OSA-HMT starches was 35.32%db and 36.36%db, respectively. This indicated that hardness is not only affected by amylose content. The adhesiveness of native starch was higher than all of the modified starches, which is in line with the other studies [47,48]. All modification treatments did not significantly affect the springiness of starch gel, but they decreased the cohesiveness, which may be due to the degradation of starch molecules, resulting in a weaker starch network structure [49]. In terms of gumminess, dual-modified starches tend to have lower values compared to both native and single-modified starch. The low level of elasticity in the dual-modified starch is influenced by its low hardness and low cohesive strength compared to single-modified starch (OSA and HMT).

### 4.6. Functional Properties

Swelling volume (SV) measures the hydration capacity of starch molecules due to the presence of water trapped in granules [50]. SV is related to amylose leaching or the solubility of starch [51]. Solubility indicates the amount of amylose leaching, which dissociates and diffuses out of the starch granules during the gelatinization process, resulting in starch swelling [10].

The SV of HMT starch decreased when compared to native starch, which is in line with other studies [2,12,52]. Furthermore, the decrease might be due to the rearrangement of starch molecules, increased intramolecular forces [52], and amylose-amylose, amylopectin-amylose, and amylopectin-amylopectin interactions, which become stronger where the starch granules become more rigid [11]. Among other modified starches, OSA starch has the highest SV. Furthermore, Park et al. [51] reported that the SV of OSA arrowroot starch was significantly higher than its native starch. This increase in swelling strength is associated with an increase in the ability of water to percolate into starch granules [53]. HMT-OSA-modified starch had a higher SV than OSA-HMT starch. It was presumed that the HMT process resulted in cracks and porous starch granules. When the OSA treatment was applied, the presence of succinate groups could weaken the internal bonds in the starch granules and increase the percolation of water so that the SV of HMT-OSA starch became higher [53,54].

All modified starches have lower solubility than native starch, and this is caused by amylose leaching during the starch gelatinization process. Amylose is in the crystalline region and is relatively small in size and linear shape, making it easier to leach out from the starch granules [25]. However, another study [24] reported that HMT on corn starch increased its solubility, which was influenced by the treatment time: the longer the treatment time, the more solubility [2].

Among the modified starches, OSA starch has the lowest WAC (0.83 g/g db). The reduced WAC on OSA starch was due to the presence of a hydrophobic substituent group from OSA that replaces the hydroxyl group on starch granules, which causes an increase in the hydrophobicity of starch [55], whereas HMT starch increased WAC (1.35 g/g db), which is in agreement with some of the previous studies [51,56]. The increased WAC of HMT starch was caused by the breaking of hydrogen bonds in the crystalline and amorphous regions, resulting in the expansion of the amorphous areas, which increased the hydrophilic properties of the starch [18,57]. The presence of pores on the surface of starch granules can also increase WAC because it will be easier for water to diffuse into the granules [12]. The increased WAC in dual-modified starch (HMT-OSA and OSA-HMT) was significantly influenced by the second modification treatment applied because HMT-OSA starch showed a lower WAC than OSA-HMT starch. OSA treatment after HMT was suspected to reduce starch hydrophilicity and increase its hydrophobicity due to the presence of OSA groups, which decreased WAC.

The esterification process with OSA increased starch’s hydrophobicity, thereby increasing the OAC [7]. On the other hand, the OAC is related to the degree of substitution (DS), where the oil absorption capacity increases with the degree of substitution of OSA [58].

Freeze-thaw stability was determined as syneresis (%). Syneresis releases water from gel or starch paste during cooling, storage, and freeze-thawing [59]. High syneresis indicates low stability at low-temperature storage [12]. Native arrowroot starch has the lowest syneresis among other starches, and this is in agreement with some of the previous studies on arrowroot starch [19,60]. All modifications applied in this study tend to increase syneresis. However, when compared with modified starches, OSA starch has higher stability, which indicates that OSA starch is more stable at low-temperature storage conditions. A previous study reported that the modification of starch into succinate derivatives (OSA starch) can improve the freeze-thaw stability (FTS) of corn and amaranth starch [50]. This is associated with a steric effect on the OSA group, which can prevent starch chain alignment when stored at low temperatures [7], whereas in this study, OSA modification cannot improve the FTS of native arrowroot starch. Increased syneresis in HMT starch might be due to the resulting random interactions reducing the water-holding capacity of the starch gel [61]. The intensity of syneresis depends on various factors, such as the composition of the amylose fraction, the length of the amylopectin chains, and the degree of polymerization of amylose and amylopectin [57].

HMT and OSA-HMT decrease SV and SOL and increase the WAC of native starch, which indicates that HMT and OSA-HMT starches can be applied in pasta and noodle formulations. Marta et al. [62] have reported that HMT banana starch can be used as an ingredient in noodle production. All modified starches have higher sineresis, so they cannot be applied in frozen foods.

### 4.7. Color Analysis

Color is a characteristic that has an important role in determining the quality and the level of consumer acceptance in selecting a product. In flour or starch products, consumers generally prefer products that are white or bright in color. To determine the color of both native and modified arrowroot starch objectively, color analysis was performed. Color testing of products, mainly arrowroot starch, can be carried out using a stand-alone, top-port color measuring instrument, which was the Spectrophotometer CM-5 in this research. The color of each starch sample was then interpreted by referring to the CIELAB systems through the L*. a*, and b* values [63].

The highest lightness (L*) was shown by OSA starch (96.39), while HMT starch showed the lowest lightness (94.96). There was a significant difference in lightness, especially for HMT-modified starch. This is presumably because thermal treatment can cause the degradation of color pigments in starch. The redness (a*) in arrowroot starch with a negative value is indicated by both native and OSA starches, which indicated a tendency to be green in color [64]. Meanwhile, the positive values for HMT starches, both single HMT and dual modified starches (HMT-OSA and OSA-HMT starches), indicated that these starches tended to be red. The color shift to red was caused by the application of thermal treatment. The level of yellowness (b*) in all arrowroot starch (native and modified) has a positive value, indicating that all starch leads to a yellow color. The highest b* value was shown by HMT starch (4.07), while the lowest was indicated by OSA starch (2.13). This means that HMT starch is more yellow than OSA starch (Figure 3).

The whiteness index (WI) is based on a scale of 0–100, with the highest value described as the highest level of lightness. The WI value is positively correlated with the L* value; the higher the WI, the higher the L* value. HMT-modified starch, for both the single HMT and dual-modified starches, had low WI. The HMT modification process that was carried out reduced the lightness level of starch. Total color difference (∆E) is a parameter used to assess how much change or difference can be seen in the Lab* value results in ingredients or food after specific treatments [65]. Dual modification, especially OSA-HMT starch, has the smallest ∆E value, which indicates that the color produced between native and OSA-HMT starch is not much different, whereas HMT starch showed the highest ∆E value, in which there was a significant difference/change in starch color compared to native starch.

## 5. Conclusions

All modification treatments, both for the single and dual modifications, had significant effects on granule morphology, crystallinity, thermal, pasting, and functional properties, texture, and color characteristics of native arrowroot starch. The granules of arrowroot starch are spherical and ellipsoid to oval shaped. There was no damage on the surface of the starch granules after OSA. In HMT-modified starch, both for single and dual modifications (HMT, HMT-OSA, and OSA-HMT), the damage occurred on the surface of the granules. Native and modified starches have a similar crystallinity pattern, which was an A-type crystalline pattern. All modification treatments decreased the RC of native starch, where HMT has a greater effect on RC than OSA. Both single HMT and dual modification (HMT-OSA, OSA-HMT) increased PT and SB, and, conversely, decreased PV and BD viscosity of native starch. All HMT treatments, both for single and dual modifications, can improve the thermal stability of native starch, especially the HMT single treatment, whereas OSA single treatment has the opposite trend compared with the other modification treatments. OSA starch has a firm texture characteristic, which is indicated by the high value of hardness and gumminess. Both single OSA treatment and HMT-OSA significantly increased the SV of native starch, but, conversely, trend for the OSA-HMT. All modified starches have lower solubility and higher syneresis than native starch. Both single HMT modification and dual modifications (HMT-OSA, OSA-HMT) increase WAC and decrease the OAC of native starch. HMT starch for both single and dual modifications have low L* and WI values, while OSA starch has the brightest color among other starches. Based on pasting and functional properties, both HMT and OSA-HMT starch can be applied in pasta and noodle formulation and also as a thickening agent, whereas OSA starch can be used as an ingredient in baby food and bakeries because it has a low ability to retrograde.

## Figures and Tables

**Figure 1 polymers-15-03215-f001:**
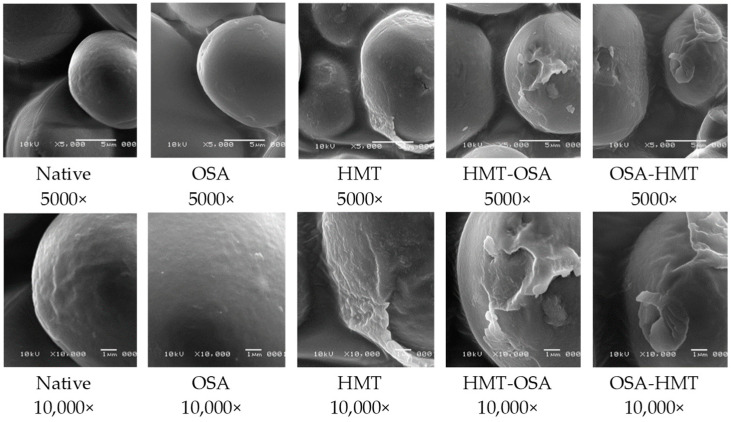
Granule morphology of native and modified arrowroot starches with 5000× and 10,000× magnification. OSA = octenyl-succinic anhydride treatment; HMT = heat-moisture treatment; HMT-OSA = HMT followed by OSA treatment; OSA-HMT = OSA followed by HMT treatment.

**Figure 2 polymers-15-03215-f002:**
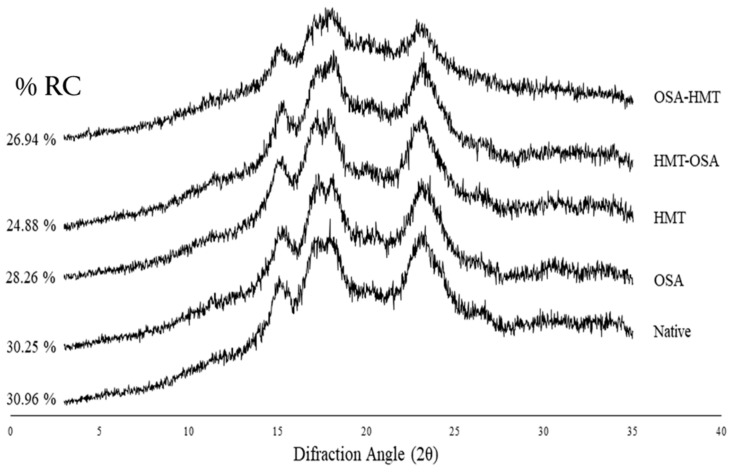
X-ray diffractograms of native and modified arrowroot starches. OSA = octenyl-succinic anhydride treatment; HMT = heat-moisture treatment; HMT-OSA = HMT followed by OSA treatment; OSA-HMT = OSA followed by HMT treatment.

**Figure 3 polymers-15-03215-f003:**
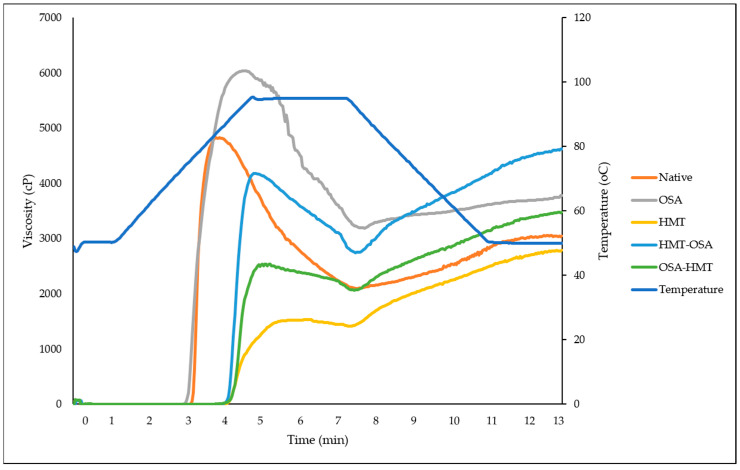
Viscoamylograph of native and modified arrowroot starches. OSA = octenyl-succinic anhydride treatment; HMT = heat-moisture treatment; HMT-OSA = HMT followed by OSA treatment; OSA-HMT = OSA followed by HMT treatment.

**Figure 4 polymers-15-03215-f004:**
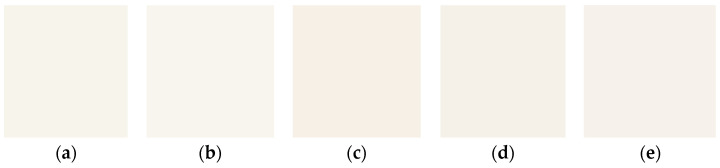
The color of arrowroot starch (**a**) native, (**b**) OSA, (**c**) HMT, (**d**) HMT-OSA, and (**e**) OSA-HMT. The color images were captured from Spectrophotometer CM-5.

**Table 1 polymers-15-03215-t001:** Thermal properties of native and modified arrowroot starches.

Treatment	T_o_ (°C)	T_p_ (°C)	T_c_ (°C)	T_c_–T_o_ (°C)	∆H (J/g)
Native	21.70 ± 3.86 ^a^	50.46 ± 5.89 ^a^	62.29 ± 3.60 ^b^	40.60 ± 7.21 ^bc^	1117.08 ± 56.09 ^a^
OSA	14.39 ± 0.69 ^b^	39.83 ± 3.96 ^b^	58.99 ± 6.67 ^bc^	44.60 ± 7.08 ^b^	647.30 ± 45.86 ^d^
HMT	20.43 ± 2.60 ^a^	44.36 ± 2.60 ^ab^	53.75 ± 1.65 ^c^	33.32 ± 2.55 ^c^	936.44 ± 35.08 ^b^
HMT-OSA	10.89 ± 3.01 ^b^	42.98 ± 3.62 ^ab^	85.76 ± 1.11 ^a^	74.87 ± 2.44 ^a^	781.64 ± 30.73 ^c^
OSA-HMT	11.38 ± 2.21 ^b^	39.98 ± 5.51 ^b^	83.96 ± 1.70 ^a^	72.58 ± 1.32 ^a^	902.26 ± 87.22 ^b^

Means marked with different letters are significantly different (*p* < 0.05). OSA = octenyl-succinic anhydride treatment; HMT = heat-moisture treatment; HMT-OSA = HMT followed by OSA treatment; OSA-HMT = OSA followed by HMT treatment.

**Table 2 polymers-15-03215-t002:** Pasting properties of native and modified arrowroot starches.

Treatment	Pasting Temperature (°C)	Peak Viscosity (cP)	Hold Viscosity (cP)	Final Viscosity (cP)	Breakdown (cP)	Setback (cP)
Native	75.74 ± 0.26 ^a^	4857 ± 42.52 ^d^	2119 ± 37.58 ^bc^	3087 ± 41.15 ^a^	2738 ± 7.00 ^d^	967 ± 15.50 ^a^
OSA	73.02 ± 0.08 ^a^	6094 ± 60.86 ^e^	3033 ± 157.88 ^d^	3956 ± 167.62 ^c^	3060 ± 216.75 ^e^	923 ± 316.74 ^a^
HMT	86.91 ± 0.42 ^c^	1604 ± 57.50 ^a^	1486 ± 60.00 ^a^	2860 ± 61.49 ^a^	118 ± 3.61 ^a^	1373 ± 9.50 ^b^
HMT-OSA	86.78 ± 0.46 ^c^	3111 ± 263.28 ^c^	2261 ± 79.00 ^c^	4081 ± 221.72 ^c^	850 ± 188.80 ^c^	1787 ± 122.11 ^c^
OSA-HMT	83.21 ± 4.03 ^b^	2544 ± 63.58 ^b^	2063 ± 99.00 ^b^	3621 ± 242.31 ^b^	481 ± 35.59 ^b^	1557 ± 166.08 ^bc^

Means marked with different letters are significantly different (*p* < 0.05). OSA = octenyl-succinic anhydride treatment; HMT = heat-moisture treatment; HMT-OSA = HMT followed by OSA treatment; OSA-HMT = OSA followed by HMT treatment.

**Table 3 polymers-15-03215-t003:** Texture profile of native and modified arrowroot starches.

Treatment	Hardness (gF)	Adhesiveness	Springiness	Cohesiveness	Gumminess
Native	244.51 ± 36.93 ^b^	−3.10 ± 2.74 ^c^	1.58 ± 1.02 ^a^	0.86 ± 0.02 ^c^	208.95 ± 27.82 ^b^
OSA	485.03 ± 35.65 ^d^	−34.20 ± 0.85 ^b^	0.90 ± 0.01 ^a^	0.76 ± 0.01 ^b^	368.29 ± 21.10 ^d^
HMT	406.78 ± 36.95 ^c^	−47.50 ± 24.71 ^b^	0.90 ± 0.04 ^a^	0.68 ± 0.05 ^a^	274.39 ± 21.43 ^c^
HMT-OSA	43.68 ± 5.89 ^a^	−54.14 ± 6.61 ^b^	0.83 ± 0.00 ^a^	0.60 ± 0.02 ^a^	29.21 ± 3.49 ^a^
OSA-HMT	76.95 ± 14.88 ^a^	−103.14 ± 6.12 ^a^	0.86 ± 0.02 ^a^	0.64 ± 0.06 ^a^	59.59 ± 10.38 ^a^

Means marked with different letters are significantly different (*p* < 0.05). OSA = octenyl-succinic anhydride treatment; HMT = heat-moisture treatment; HMT-OSA = HMT followed by OSA treatment; OSA-HMT = OSA followed by HMT treatment.

**Table 4 polymers-15-03215-t004:** Functional properties of native and modified arrowroot starches.

Treatment	Swelling Volume (mL/g db)	Solubility (%)	WAC (g/g db)	OAC (g/g db)	Syneresis (%)
Native	16.27 ± 0.12 ^c^	9.33 ± 0.14 ^c^	1.13 ± 0.20 ^b^	2.33 ± 0.03 ^b^	7.02 ± 0.06 ^a^
OSA	25.84 ± 0.56 ^e^	9.13 ± 0.21 ^b^	0.83 ± 0.10 ^a^	2.31 ± 0.07 ^b^	18.71 ± 0.96 ^b^
HMT	11.46 ± 0.54 ^a^	8.92 ± 0.14 ^a^	1.35 ± 0.05 ^c^	2.15 ± 0.12 ^a^	38.76 ± 0.49 ^c^
HMT-OSA	17.66 ± 0.17 ^d^	9.08 ± 0.09 ^b^	1.41 ± 0.07 ^c^	2.21 ± 0.09 ^a^	59.06 ± 2.27 ^d^
OSA-HMT	14.31 ± 0.34 ^b^	8.90 ± 0.04 ^a^	1.60 ± 0.06 ^d^	2.20 ± 0.04 ^a^	65.80 ± 3.54 ^e^

Means marked with different letters are significantly different (*p* < 0.05). OSA = octenyl-succinic anhydride treatment; HMT = heat-moisture treatment; HMT-OSA = HMT followed by OSA treatment; OSA-HMT = OSA followed by HMT treatment.

**Table 5 polymers-15-03215-t005:** Color parameters of native and modified arrowroot starches.

Treatment	L*	a*	b*	Whiteness Index	∆E
Native	96.26 ± 0.03 ^d^	−0.25 ± 0.01 ^a^	3.18 ± 0.03 ^c^	95.09 ± 0.01 ^d^	-
OSA	96.39 ± 0.19 ^d^	−0.29 ± 0.01 ^a^	2.13. ± 0.10 ^a^	95.80 ± 0.11 ^e^	1.08 ± 0.11
HMT	94.96 ± 0.12 ^a^	0.09 ± 0.00 ^b^	4.07 ± 0.01 ^d^	93.52 ± 0.09 ^a^	1.61 ± 0.09
HMT-OSA	95.44 ± 0.24 ^b^	0.05 ± 0.03 ^b^	3.07 ± 0.23 ^bc^	94.51 ± 0.33 ^b^	0.92 ± 0.13
OSA-HMT	95.80 ± 0.09 ^c^	0.07 ± 0.07 ^b^	2.97 ± 0.25 ^b^	94.86 ± 0.22 ^c^	0.64 ± 0.09

Means marked with different letters are significantly different (*p* < 0.05). OSA = octenyl-succinic anhydride treatment; HMT = heat-moisture treatment; HMT-OSA = HMT followed by OSA treatment; OSA-HMT = OSA followed by HMT treatment.

## Data Availability

Not applicable.

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
