# Peer review of "The Effect of Dual-Modification by Heat-Moisture Treatment and Octenylsuccinylation on Physicochemical and Pasting Properties of Arrowroot Starch"

_polymers, 2023, doi:10.3390/polym15153215_

Round 1

Reviewer 1 Report

83. Please mention the procedure for precipitation. Did you mean incubation or stirring of the slurry by precipitation?

86. The precipitation from 18hr and followed by centrifugation, whether this process repeated 3-4 times? If so the isolation of starch will be time-consuming but previous studies on arrowroot starch have less time-consuming procedures then, what is the possible explanation for this isolation procedure you have chosen?

87. Usually starch slurry is used to dry around 40 to 45°Cin a hot air oven if you are keeping it for two days. Here you have abovementioned in the introduction that native starch has low thermal stability, then what is the reason for keeping the starch at 50°C?

104. pH is neutralized means ph 7. Here you have mentioned 6.5, so it is better to remove the term ‘neutralized’ from this sentence.

193. ‘w’ in the equation stands for weight or volume?

210.  where is the equation for the whiteness index of the native starch?

Fig 1. Align the figure with respect to the text. Add legends below the title

235. Where are the RC values for the native and modified starches?

Figure 2. Add legends to the title.

243. Explain the term ‘native of the starch gelatinization’.

253- Reframe the sentence. You have mentioned the enthalpy decreased after modification from 45.83 to 47.80°C which is the temperature range of native and OSA-HMT respectively. Whereas the next range. Here nowhere in the table mentioned the value 67.50°c.

Table 1. Insert the legend below the table and Means marked with different letters. Explain the higher standard deviation values observed for ΔH for the starches.

Figure 3. Insert the legend below the title. Check the font and font size of the axis titles.

3.7. Mention the lightness index ΔE in the results.

345. Mention the RC values of native and modified samples in both Results and Discussion parts.

367. Add reference.

378. Add a reference for your justification.

383. Add year to the citation

406. What is the amylose content of the native and modified starch samples?

417. Insert citation for your explanation

427-429. Reframe the sentence.

494. Insert citation for your justification

4.6 Functional properties:- Include the possible industrial application of the improved functional properties after modification likewise alter the conclusion

The English language used to write the manuscript can be improved. Some sentences mentioned in the comments need to be reframed with proper grammar checking.

Author Response

Responds to the reviewers’ comments:

Thank you for your comments concerning our manuscript entitled “The Effect of Dual-Modification by Heat-Moisture Treatment and Octenylsuccinylation on Physicochemical and Pasting Properties of Arrowroot Starch”. Those comments are valuable and very helpful for revising and improving our paper. We have studied the comments carefully and made a correction that we hope meets with approval. Revised portions are marked in yellow on the paper. The main correction and the responses to the reviewer’s comment are as follows:

Reviewer 1

  1. Please mention the procedure for precipitation. Did you mean incubation or stirring of the slurry by precipitation?

Response:

precipitation means to leave the starch solution for 18 hours without stirring until starch precipitates are formed and then decanted to separate precipitated starch from the supernatant (Lines 84-89)

  1. The precipitation from 18hr and followed by centrifugation, whether this process repeated 3-4 times? If so the isolation of starch will be time-consuming but previous studies on arrowroot starch have less time-consuming procedures then, what is the possible explanation for this isolation procedure you have chosen?

Response:

The first precipitation was done manually which was allowed to settle for 18 hours. The precipitated starch was decanted to separate starch and supernatant. The resulting starch was washed and then precipitation again using a centrifuge. The washing process was carried out 3 times until the water was clear. The separation process using a centrifuge only takes 2 minutes per run, so it is not time-consuming (Lines 87-89)

  1. Usually starch slurry is used to dry around 40 to 45°Cin a hot air oven if you are keeping it for two days. Here you have above mentioned in the introduction that native starch has low thermal stability, then what is the reason for keeping the starch at 50°C?

Response:

We use a drying temperature of 50°C for 24 hours for drying the starch. The statement "starch has low thermal stability" refers to the characteristics of starch when gelatinized, not in the form of dry starch. The "starch has low thermal stability" means the starch gel reduces its viscosity after reaching a peak viscosity in the gelatinization process.

  1. pH is neutralized means ph 7. Here you have mentioned 6.5, so it is better to remove the term ‘neutralized’ from this sentence.

Response:

The word “neutralized” was replaced with “adjusted” (Line 106)

  1. ‘w’ in the equation stands for weight or volume?

Response:

“w” stands for weight. It has been revised in Equation 5 (Line 195)

  1. where is the equation for the whiteness index of the native starch?

Response:

the whiteness index equation for native starch and modified starch are the same (equation 7) (Line 207)

Fig 1. Align the figure with respect to the text. Add legends below the title

Response:

It has been revised (Lines 231-234)

  1. Where are the RC values for the native and modified starches?

Response:

The RC value of native and modified starch are presented on the left side in Figure 2.

Figure 2. Add legends to the title.

Response:

It has been added (Lines 248-250)

  1. Explain the term ‘native of the starch gelatinization’.

Response:

It has been revised. The sentence has been rearranged (Lines 253-254)

253- Reframe the sentence. You have mentioned the enthalpy decreased after modification from 45.83 to 47.80°C which is the temperature range of native and OSA-HMT respectively. Whereas the next range. Here nowhere in the table mentioned the value 67.50°c.

Response:

It has been revised. The sentence has been rearranged (Lines 266-268)

Table 1. Insert the legend below the table and Means marked with different letters.

Response:

It has been inserted and added (Table 1, lines 270-272)

Explain the higher standard deviation values observed for ΔH for the starches.

Response:

Thank you for your correction. We have re-analyzed and obtained data as in Table 1.

Figure 3. Insert the legend below the title. Check the font and font size of the axis titles.

Response:

It has been revised and added (Figure 3, Lines 281-283)

3.7. Mention the lightness index ΔE in the results.

Response:

It has been mentioned (Lines 328-333)

  1. Mention the RC values of native and modified samples in both Results and Discussion parts.

Response:

It has been mentioned (Lines 241-244; 368-381)

  1. Add reference.

Response:

It has been added (Line 391)

  1. Add a reference for your justification.

Response:

It has been added (Line 406)

  1. Add year to the citation

Response:

We have followed the journal template. Based on the journal template, the citation at the beginning of the sentences, we can write just the last name of the first author without the year (Line 407)

  1. What is the amylose content of the native and modified starch samples?

Response:

It has been added in the paragraph (Lines 437-449)

Sample

Amylose Content (% db)

Native

29,15 ± 2,71a

HMT

36,01 ± 1,49c

OSA

33,44 ± 3,06b

HMT-OSA

35,32 ± 0,64bc

OSA-HMT

36,36 ± 1,70c

  1. Insert citation for your explanation

Response:

It has been inserted (Line 453)

427-429. Reframe the sentence.

Response:

It has been reframed (Line 461-463)

  1. Insert citation for your justification

Response:

It has been inserted (Lines 535-537)

4.6 Functional properties: - Include the possible industrial application of the improved functional properties after modification likewise alter the conclusion

Response:

It has been included (Lines 424-431; 517-521)

Reviewer 2 Report

Why were the parameters related to the retrogradation process not marked in the DSC analysis? Then %R can be calculated. I suggest doing it.

Author Response

Reviewer 2

Why were the parameters related to the retrogradation process not marked in the DSC analysis? Then %R can be calculated. I suggest doing it.

Response:

Thank you for your valuable suggestion. It is very interesting. But in this study, we did not determine the sample after storage at low temperatures for several days, so we can not calculate the %R. We have limited funding to carry out the analysis. We hope in the next study, we will do it.

According to Wang, et al. [1], the degree of retrogradation (%R) was calculated according to the formula:

Where:

∆Hg = gelatinization enthalpy changes of native starch

∆Hr = Enthalpy change on reheating of retrograded starch gels

Starch retrogradation was determined on the same gelatinized samples after storage at 4 °C for 7 days.

  1. Wang, S.; Li, C.;  Zhang, X.;  Copeland, L.; Wang, S. Retrogradation Enthalpy Does Not Always Reflect the Retrogradation Behavior of Gelatinized Starch. Scientific Reports 2016, 6, 20965.

Round 2

Reviewer 1 Report

Line 88-92. The sentences are repeated. Rephrase accordingly.

line 225. Add the term "treatment" after OSA.

4.1 . Mention any previous research on starch having similar observations

Author Response

Responds to the reviewers’ comments:

Thank you for your comments concerning our manuscript entitled “The Effect of Dual-Modification by Heat-Moisture Treatment and Octenylsuccinylation on Physicochemical and Pasting Properties of Arrowroot Starch”. Those comments are valuable and very helpful for revising and improving our paper. We have studied the comments carefully and have made a correction which we hope meets with approval. Revised portions are marked in blue on the paper. The main correction and the responses to the reviewer’s comment are as follows:

Reviewer 1

Line 88-92. The sentences are repeated. Rephrase accordingly.

Response:

It has been rephrased (Lines 88-90)

line 225. Add the term "treatment" after OSA.

Response:

It has been added (Line 224)

4.1. Mention any previous research on starch having similar observations

Response:

It has been mentioned (Line 334; 346-347; 349)

Reviewer 2 Report

The manuscript was improved according to Reviewer sugestions. 

Author Response

Thank you for your comments